physical chemistry/optics

terahertz, ice, sublimation, phase change

**Author for correspondence:**
Kei Takeya
e-mail: takeya@nuee.nagoya-u.ac.jp

# Observation of sublimation of ice using terahertz spectroscopy

Keisuke Matsumura, Kodo Kawase and Kei Takeya

Department of Electronics, Nagoya University, Nagoya, Japan

KT, 0000-0003-3055-5288

Although many studies have investigated the phase change of water, few have focused on the sublimation of ice. This study revealed that ice sublimation can be observed using terahertz (THz) spectroscopy. From measurements in the range of 210–270 K, the sublimation was observed over the entire temperature range and the rate of sublimation was increased proportionally with temperature. Particularly on a time scale of a few hundred minutes, the sublimation progresses visibly above 250 K. Above a certain temperature, the absorption coefficient increased during sublimation. These findings suggest that an interesting phenomenon may occur in the phase change of water at sub-zero temperatures, indicating that THz spectroscopy would be useful for measuring water and ice.

## 1. Introduction

Water has been investigated in various ways [1,2]. However, compared to studies of phase changes in water, such as evaporation, melting and freezing, relatively few studies have focused on sublimation [3]. Sublimation is also a phase change; it is closely related to climate change in polar regions and snow-covered areas [4], and to the existence of ice on celestial bodies [5]; it is also important in food science [6]. Sublimation occurs at the ice surface, and the quasi-liquid layer that forms during surface phase changes below the freezing point has been discussed for a long time [1,7,8]. Previous reports on sublimation have been mainly based on the analysis of mass changes associated with temperature and pressure changes [9–11]. However, direct observation of the sublimation process has only been achieved within the past decade [12,13]. This is because it is difficult to distinguish between ice and water at the ice surface. This study demonstrates the possibility of novel observations of ice sublimation.

Terahertz (THz) wave spectroscopy was used for measurements due to the characteristic response of THz waves to water [14–21]. THz waves have moderate transmission through solid ice [2,15–17], but they are strongly absorbed by liquid water [1,18–20] and have a fingerprint spectrum in water vapour [21], so THz

spectroscopy can be used to distinguish between the three phases of water. Additionally, both real and imaginary parts of the optical parameters of a measured object can be determined using THz-time domain spectroscopy (TDS) [14]. The real parameter represents the optical distance, which is the thickness of the sample, and the imaginary parameter represents the absorbance of the sample. Therefore, we examined whether sublimation could be observed using THz-TDS.

## 2. Experimental

Solid ice was prepared from ultrapure water at a temperature of 263 K using an environmental tester (SU-221; Espec Corp., Osaka, Japan). The prepared ice was crushed and powdered using a mortar and then formed into tablets 10 mm in diameter and about 1–2 mm thick using a tablet press (HANDTAB-Jr.; Ichihashi Seiki, Kyoto, Japan) in a cryogenic freezer (VT-208; Nippon Freezer Co., Ltd, Saitama, Japan).

The ice tablets were measured using a THz-TDS system with a femtosecond fibre laser (IMRA, $\lambda =$ 780 nm, pulse width: 100 fs, repetition frequency: 50 MHz) as an excitation light source (figure 1). A photoconductive antenna was used for both generation and detection of THz waves. The complex optical constant of the sample was calculated from the acquired time domain waveform. The details of the procedure were essentially the same as described previously [16,22]. In addition, the THz optical path was purged with nitrogen to reduce the influence of water vapour in the air. Ice samples were kept at atmospheric pressure in a nitrogen gas atmosphere during measurement.

In addition to the above THz-TDS system, measurements were performed using a refrigerator system (POGT-205D; Pascal, Osaka, Japan/RDK-101D; Sumitomo Heavy Industries, Ltd, Tokyo, Japan) to control the temperature of the object. The internal temperature could be changed from 10 K to about room temperature using two heaters installed inside the system and lowering the temperature with the circulation of helium gas inside the cooler. The temperature accuracy of the experiment was ±0.1 K. The optical parameter error was 0.5% based on the thickness measurements.

The complex optical parameters of the sample, $n(\omega)$ and $k(\omega)$, are calculated from the following equations:

$$\frac{E_{\text{sam}}(\omega)}{E_{\text{ref}}(\omega)} = \rho(\omega)\exp[-i\varphi(\omega)],\tag{2.1}$$

$$n(\omega) = \frac{c\varphi(\omega)}{\omega L} + 1\tag{2.2}$$

and
$$k(\omega) = \frac{c}{\omega L}\ln\left\{\frac{4n(\omega)}{\rho(\omega)\,[n(\omega)+1]^2}\right\},\tag{2.3}$$

where $\omega$ is the frequency, $E_{\text{sam}}(\omega)$ is the fast Fourier transform (FFT) spectrum of the THz pulse propagating through the samples, $E_{\text{ref}}(\omega)$ is the reference FFT spectrum, $\rho(\omega)$ is the transmission, $\phi(\omega)$ is the phase shift, $c$ is the light speed and $L$ is the thickness of the sample [23].

## 3. Results and discussion

The time domain waveform transmitted through the ice sample shifted with time, as shown in figure 2, when ice was placed at atmospheric pressure and 260 K and observed with THz-TDS. The position of the time domain waveform observed by THz-TDS is dependent on the optical distance $d$, and this time domain waveform shifts due to changes in the refractive index $n$ or the thickness $t$ of the sample from the relationship of $\Delta d = \Delta n * \Delta t$. Under these temperature and pressure conditions, there can be no large refractive index changes in the substances produced by water [15–20]. Therefore, the changes seen here were not changes in the refractive index, but seemed to indicate changes in thickness. When the actual thickness was measured and compared to the time domain waveform, a correlation was observed between the measured change in thickness of the sample held for about 180 min at 260 K and the shift in the time domain waveform, as shown in figure 3. Therefore, shifts in the time domain waveform could be used as an indicator of changes in the thickness of the measured sample.

The thinning of the ice was indicative of a decrease in volume and represented sublimation. Therefore, sublimation of solid crystals could be observed without contact from the time domain waveform of THz-TDS.

Next, the temperature dependence of the time domain waveform shift was measured. The shift in the time domain waveform was measured at increments of 10 K between 210 and 270 K. Here, the observed shift corresponded to a change in thickness. The results are shown in figure 4. The change in sample

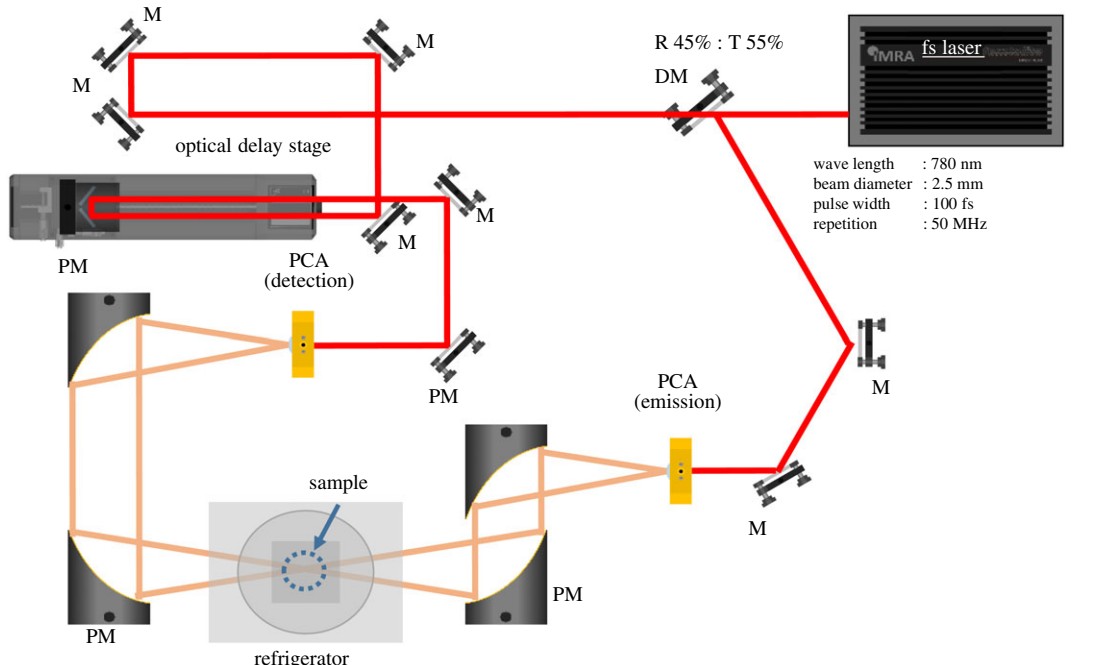

**Figure 1.** Schematic diagram of THz-TDS. M, mirror; DM, dichroic mirror; PM, parabolic mirror; R, reflectance; T, transmittance; PCA, photoconductive antenna.

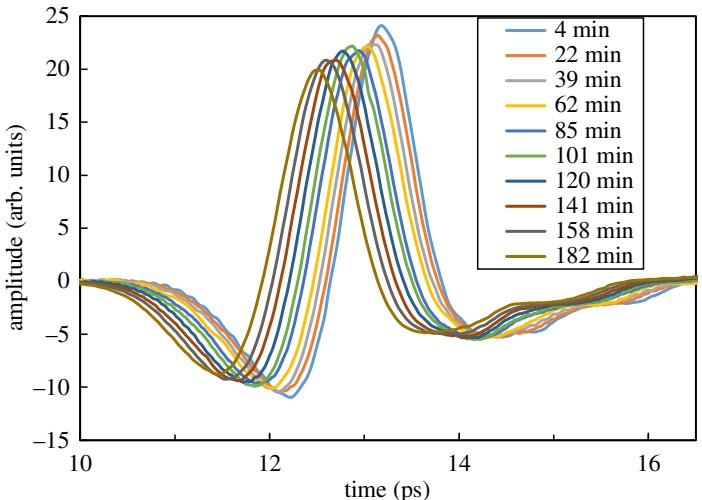

**Figure 2.** Changes in the time domain waveform were observed in ice at atmospheric pressure and at 260 K. The waveform shifted to the left (in the direction of shortening of the optical path length) over time.

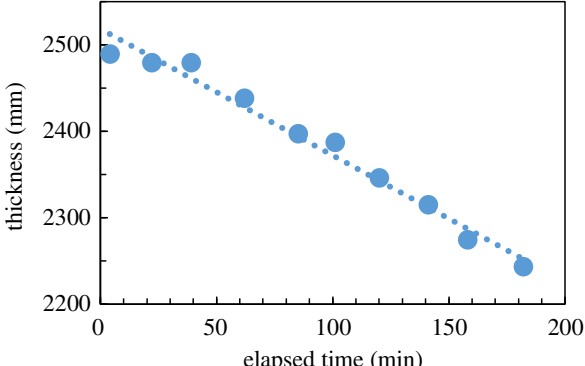

**Figure 3.** Correlation between shift in the time domain waveform and sample thickness.

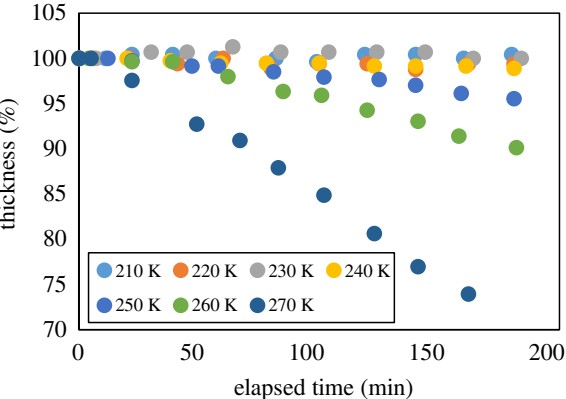

**Figure 4.** Shift in the time domain waveform as a function of thickness and temperature (ranging from 210 to 270 K). The slope becomes steeper, indicating increasing sublimation rate, at higher temperatures.

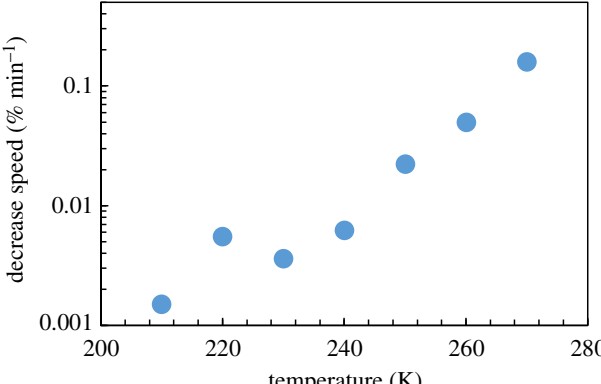

**Figure 5.** Temperature dependence of the sublimation rate.

thickness was linearly related to the elapsed time and the slope became steeper with increasing temperature. This indicates that the speed of sublimation increased with increasing temperature. As shown in figure 4, the slope was gentle up to 240 K, and then became steep above this temperature. This linear relation as a function of elapsed time seemed to be due to the tablet shape of the sample. In the case of sublimation, the speed was proportional to the surface area of the tablet. Therefore, although sublimation proceeded, the surface area was nearly constant and isostatic.

Figure 5 presents the change in slope (Δthickness/Δelapsed time; see figure 4) as a function of temperature. It is observed that the decrease speed of the ice sample volume increases with the rise in temperature. The temperature dependence is almost linear on the logarithmic scale. The behaviour of this temperature dependence is consistent with the previous papers on vapour pressure measurements during sublimation [9–11]. Although the observed physical quantities are different from those in the previous papers, it is found that the response to temperature is similar. This indicates the validity of our experimental results. Further, on the measurement time scale in the present study, a significant decrease of thickness was observed above 250 K.

The complex optical parameters of the sublimating ice were calculated considering the changes in thickness. Figure 6 presents the transition of the refractive index, and figure 7 presents the transition of the absorption coefficient. The values varied according to temperature, reflecting the temperature dependence of the refractive index of the hydrogen-bonding solid crystal [16]. The refractive index of materials such as ice increases with temperature [16], but changes in the values of the refractive index with respect to time changes were small at all temperatures. This calculation method considering the changing thickness was correct because the optical constant was almost constant. The refractive index indicates that no significant changes to the ice occurred.

By contrast, the absorption coefficient increased with time at temperatures over 240 K. The temperature range over which this increase was observed coincided with the temperature range in

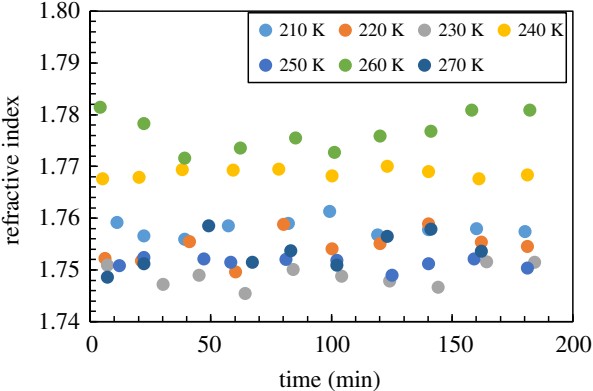

**Figure 6.** The refractive index of ice as a function of time and temperature. The index remained relatively constant as a function of temperature.

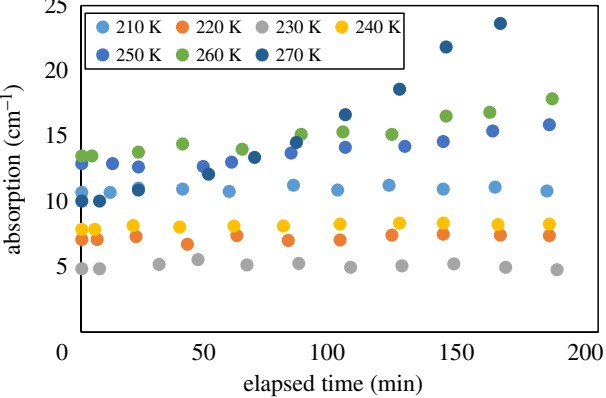

**Figure 7.** The absorption coefficient of ice as a function of time and temperature.

which sublimation was observed, indicating that some phenomenon was at play. For example, various studies have shown that there is a quasi-liquid phase in the phase transition of the surface of ice below the freezing point [8,12,13]. The quasi-liquid phase is a phenomenon that has been predicted for a long time, but was difficult to observe until recent studies using microscopy. In the present study, THz waves were shown to pass through ice, while they were strongly absorbed by liquid water, making them a useful tool for identifying ice and water. The absorption coefficients of water and ice at 1 THz are 240 and 7 cm$^{-1}$, respectively, and the difference between them is about 35 times [16–19]. Because ice and water vapour do not strongly absorb THz waves [15–21], the increase in the absorption coefficient observed here may have been due to the presence of liquid water. The increase in absorption was observed above 240 K [24,25], which is close to the limit of temperature for the presence of supercooled water at atmospheric pressure. The observed increase in absorption appeared to be related to liquid water, so an interesting phenomenon in the quasi-liquid phase or supercooled water—sublimation—could be observed. Here, supercooled water was generated only during the process of supercooling. Thus, the observation of supercooled water in this study indicates that liquid water is generated at any temperature. It is likely that liquid water is frozen immediately after generation, becoming observable at 240 K. Another possibility is that metastable water is generated, which has never been reported previously. The refractive indices of ice and water in the THz frequency are approximately 1.8 and 2.2, with a difference of approximately Δ0.4 [13,15]. This difference is not so small as to be negligible. However, if liquid water is present on the ice surface during sublimation, it is probably only on the surface of the ice sample and the amount seems to be very small. Therefore, it does not have a large effect on the refractive index measured in this experiment, but mainly has a large effect on absorption only. This result is consistent with the results of the present experiment, where only absorption is increased, and supports the hypothesis of the presence of water. These observations revealed the sublimation of ice, which cannot be observed using other methods, and this interesting phenomenon will be of great interest in both the material science of ice and THz wave science.

# 4. Conclusion

We demonstrated that THz spectroscopy can be used to observe the sublimation of ice. At atmospheric pressure and temperatures below freezing, the sublimation of ice was observed from the shift in the time domain waveform. The sublimation is observed in the entire measurement temperature range 210–270 K, and the rate is proportional to the temperature. The sublimation progresses visibly above 250 K. Additionally, the absorption coefficient increased during sublimation. These observations may indicate an interesting phenomenon in the phase change of ice, and will open new perspectives in material science studies of water.

Ethics. This article does not present research with ethical considerations.

Data accessibility. Data available from the Dryad Digital Repository: https://dx.doi.org/10.5061/dryad.fttdz08nz [26].

Authors' contributions. K.T. conceived the idea of the ice sublimation. All authors designed the experiment, K.M. and K.T. performed the THz-TDS experiments, K.T. is the corresponding author, K.M. and K.T. wrote the main manuscript text and K.K. is the supervisor of this work. All authors reviewed the manuscript.

Competing interests. We declare we have no competing interests.

Funding. This work was supported by the JSPS KAKENHI (grant nos. 23760846, 25709091 and 17H03535). We thank Casio Science Promotion Foundation and Shimadzu Co. for financial assistance throughout this investigation.

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
