## [Reviewer comments · Royal Society Open Science]

Review History

RSOS-192083.R0 (Original submission)

Review form: Reviewer 1

Is the manuscript scientifically sound in its present form?

No

Are the interpretations and conclusions justified by the results?

No

Is the language acceptable?

Yes

Do you have any ethical concerns with this paper?

No

Have you any concerns about statistical analyses in this paper?

No

Recommendation?

Major revision is needed (please make suggestions in comments)

Comments to the Author(s)

In this manuscript by Matsumura et al., the authors measured the sublimation rates of ice as a function of temperature. These data were obtained by two different methods. To date, however, there are so many papers reported on the sublimation of ice. If the authors can provide detailed comparisons with the results and reported values, the reliability of this study may be strength. On the other hand, the advances of this paper are the use of THz for the sublimation. If the authors can put more detailed discussions on the sublimation based on above comments, this paper could be worth publishing to RSOS.

Review form: Reviewer 2

Is the manuscript scientifically sound in its present form?

Yes

Are the interpretations and conclusions justified by the results?

Yes

Is the language acceptable?

Yes

Do you have any ethical concerns with this paper?

No

Have you any concerns about statistical analyses in this paper?

No

Recommendation?

Major revision is needed (please make suggestions in comments)

Comments to the Author(s)

Journal: Royal Society Open Science

Manuscript Number: RSOS-192083

Title: "Observation of Sublimation of Ice using Terahertz Spectroscopy"

Authors: Matsumura, Keisuke; Kawase, Kodo; Takeya, Kei

Recommendation: The manuscript could be considered for publication after some major modifications.

Reviewing report:

Water and/or Ice is vital for nature and some intrinsic physical mechanism cannot be well understood from the system. Spectroscopy is a good method to check the composition and molecule vibration, especially for the infrared wavelength region. Due to the limitations from instrument, terahertz (THz) (far-infrared) properties from Water and/or Ice have not been well discovered, even there were some reports. In the manuscript, the authors reported the ice sublimation process by using THz spectroscopy. It was found that Ice undergone sublimation at atmospheric pressure at temperatures above 240 K and the absorption coefficient or sublimation velocity can be correspondingly changed. In my opinion, these results are interesting and can be accepted for publication. I would like to recommend the manuscript to be accepted after some modifications according to the comments as following:

- 1) THz experimental system should be presented in order to understand the scientific process.
- 2) A detailed comparison of refractive index in THz region between Water and Ice during the sublimation process should be presented and discussed in details.
- 3) If possible, how the THz data can derive the optical constants should be added.

4) In addition, all figures are needed to be improved for the publication level.

Decision letter (RSOS-192083.R0)

Dear Dr Takeya:

Title: Observation of Sublimation of Ice using Terahertz Spectroscopy
Manuscript ID: RSOS-192083

Thank you for your submission to Royal Society Open Science. The chemistry content of Royal Society Open Science is published in collaboration with the Royal Society of Chemistry. I am very sorry it has taken much longer than usual for us to be able to send you a decision on your manuscript.

The editor assigned to your manuscript has now received comments from reviewers. We would like you to revise your paper in accordance with the referee and Subject Editor suggestions which can be found below (not including confidential reports to the Editor). Please note this decision does not guarantee eventual acceptance.

Please submit your revised paper before 15-Jul-2020. Please note that the revision deadline will expire at 00.00am on this date. If we do not hear from you within this time then it will be assumed that the paper has been withdrawn. In exceptional circumstances, extensions may be possible if agreed with the Editorial Office in advance. We do not allow multiple rounds of revision so we urge you to make every effort to fully address all of the comments at this stage. If deemed necessary by the Editors, your manuscript will be sent back to one or more of the original reviewers for assessment. If the original reviewers are not available we may invite new reviewers.

Royal Society of Chemistry
Thomas Graham House

Science Park, Milton Road
Cambridge, CB4 0WF
Royal Society Open Science - Chemistry Editorial Office

On behalf of the Subject Editor Professor Anthony Stace and the Associate Editor Professor Tobias Hertel.

RSC Associate Editor:
Comments to the Author:
(There are no comments.)

RSC Subject Editor:
Comments to the Author:
(There are no comments.)

Reviewers' Comments to Author:

Reviewer: 1

Comments to the Author(s)

In this manuscript by Matsumura et al., the authors measured the sublimation rates of ice as a function of temperature. These data were obtained by two different methods. To date, however, there are so many papers reported on the sublimation of ice. If the authors can provide detailed comparisons with the results and reported values, the reliability of this study may be strength. On the other hand, the advances of this paper are the use of THz for the sublimation. If the authors can put more detailed discussions on the sublimation based on above comments, this paper could be worth publishing to RSOS.

Reviewer: 2

Comments to the Author(s)

Journal: Royal Society Open Science

Manuscript Number: RSOS-192083

Title: "Observation of Sublimation of Ice using Terahertz Spectroscopy"

Authors: Matsumura, Keisuke; Kawase, Kodo; Takeya, Kei

Recommendation: The manuscript could be considered for publication after some major modifications.

Reviewing report:

Water and/or Ice is vital for nature and some intrinsic physical mechanism cannot be well understood from the system. Spectroscopy is a good method to check the composition and molecule vibration, especially for the infrared wavelength region. Due to the limitations from instrument, terahertz (THz) (far-infrared) properties from Water and/or Ice have not been well discovered, even there were some reports. In the manuscript, the authors reported the ice sublimation process by using THz spectroscopy. It was found that Ice undergone sublimation at atmospheric pressure at temperatures above 240 K and the absorption coefficient or sublimation velocity can be correspondingly changed. In my opinion, these results are interesting and can be accepted for publication. I would like to recommend the manuscript to be accepted after some modifications according to the comments as following:

- 1) THz experimental system should be presented in order to understand the scientific process.
- 2) A detailed comparison of refractive index in THz region between Water and Ice during the sublimation process should be presented and discussed in details.
- 3) If possible, how the THz data can derive the optical constants should be added.

4) In addition, all figures are needed to be improved for the publication level.

Author's Response to Decision Letter for (RSOS-192083.R0)

See Appendices A & B.

Decision letter (RSOS-192083.R1)

Dear Dr Takeya:

Title: Observation of Sublimation of Ice using Terahertz Spectroscopy
Manuscript ID: RSOS-192083.R1

It is a pleasure to accept your manuscript in its current form for publication in Royal Society Open Science. The chemistry content of Royal Society Open Science is published in collaboration with the Royal Society of Chemistry.

On behalf of the Subject Editor Professor Anthony Stace and the Associate Editor Professor Tobias Hertel.

RSC Associate Editor
Comments to the Author:
(There are no comments.)

Reviewer(s)' Comments to Author:

Appendix A

Reply for the reviewer 1:

Thank you for your useful and constructive comments. Taking the reviewers' comment into account, we have revised the manuscript and provided replies to the reviewers' comment (in bold).

Reviewers' Comments to Author:

Reviewer: 1

Comments to the Author(s)

In this manuscript by Matsumura et al., the authors measured the sublimation rates of ice as a function of temperature. These data were obtained by two different methods. To date, however, there are so many papers reported on the sublimation of ice. If the authors can provide detailed comparisons with the results and reported values, the reliability of this study may be strength. On the other hand, the advances of this paper are the use of THz for the sublimation. If the authors can put more detailed discussions on the sublimation based on above comments, this paper could be worth publishing to RSOS.

Thank you for reading our paper. In this paper, we have used terahertz spectroscopy as a new method to observe ice sublimation. Several previous papers on the sublimation have reported on vapor pressure studies. These methods have been carried out by precise mass spectrometry. Therefore, all the changes that occur in sublimation are observed as changes in mass. In this process, pressure-temperature measurements have been made to measure the vapor pressure.

Our method, on the other hand, takes a completely different approach than previously reported. We conduct our experiments by measuring the temporal waveforms of electromagnetic waves transmitted through the ice sample, and the phase delay and absorption of the THz waves can be measured. From these results, the complex optical constant of the sample can be determined. By measuring the phase delay of the THz, the volume and shape changes of the sample can be measured, and the changes in absorbance can also provide information on the phase changes of the ice. Thus, our method provides new information compared to previous experiments and shows that new analyses can be performed.

Comparison of this paper with previous papers is somewhat difficult due to the different information obtained by them. However, from detailed analysis of the experimental results of the sublimation speed with respect to temperature, we find that the sublimation speed is

linear on a logarithmic scale. This result is consistent with the results found in previous papers, (R. Feistel, W. Wagner. *Geochim. Cosmochim. Acta*, 2007, 71, 36. C. E. Bryson III, V. Cazcarra, L. L. Levenson *J. Chem. Eng. Data*, 1974, 19, 107. J. Marti, K. Mauersberger. *Geophys. Res. Lett.*, 1993, 20, 363.) and we believe that it shows the validity of our experimental results.

Based on the above comments, we have added citations, made additions and revised figures in the paper. A paper highlighting the revisions is also attached. Please check it out.

Appendix B

Reply for the reviewer 2:

Thank you for your useful and constructive comments. Taking the reviewers' comment into account, we have revised the manuscript and provided replies to the reviewers' comment (in bold).

Reviewer: 2

Reviewing report:

Water and/or Ice is vital for nature and some intrinsic physical mechanism cannot be well understood from the system. Spectroscopy is a good method to check the composition and molecule vibration, especially for the infrared wavelength region. Due to the limitations from instrument, terahertz (THz) (far-infrared) properties from Water and/or Ice have not been well discovered, even there were some reports. In the manuscript, the authors reported the ice sublimation process by using THz spectroscopy. It was found that Ice undergone sublimation at atmospheric pressure at temperatures above 240 K and the absorption coefficient or sublimation velocity can be correspondingly changed. In my opinion, these results are interesting and can be accepted for publication. I would like to recommend the manuscript to be accepted after some modifications according to the comments as following:

- 1) THz experimental system should be presented in order to understand the scientific process.
- 2) A detailed comparison of refractive index in THz region between Water and Ice during the sublimation process should be presented and discussed in details.
- 3) If possible, how the THz data can derive the optical constants should be added.
- 4) In addition, all figures are needed to be improved for the publication level.

Thank you for reading our paper. We hope that THz spectroscopy will develop as a new observation tool for the sublimation, and thank you for understanding the purpose of our paper. The responses to your comments are listed below.

1) A diagram of the terahertz experimental system has been added as figure 1.

2) In the main text, we have added the following sentences,

“The refractive indices of ice and water in the THz frequency are approximately 1.8 and 2.2, with a difference of approximately $\Delta 0.4$.^{13, 15} This difference is not so small as to be negligible. However, if liquid water is present on the ice surface during sublimation, it is probably only on the surface of the ice sample and the amount seems to be very small. Therefore, it does not have large effect on the refractive index measured in this experiment,

but mainly has a large effect on absorption only. This result is consistent with the results of the present experiment, where only absorption is increased, and supports the hypothesis of the presence of water.

3) The following text was added to the experiment section.

“The complex optical parameters of the sample, $n(\omega)$ and $k(\omega)$, are calculated from the following equations

$$\frac{E_{sam}(\omega)}{E_{ref}(\omega)} = \rho(\omega) \exp[-i\phi(\omega)] \quad (1)$$

$$n(\omega) = \frac{c\phi(\omega)}{\omega L} + 1 \quad (2)$$

$$k(\omega) = \frac{c}{\omega L} \ln \left\{ \frac{4n(\omega)}{\rho(\omega)[n(\omega)+1]^2} \right\} \quad (3)$$

where ω is the frequency, $E_{sam}(\omega)$ is the fast Fourier transform (FFT) spectrum of the THz pulse propagating through the samples, $E_{ref}(\omega)$ is the reference FFT spectrum, $\rho(\omega)$ is the transmission, $\phi(\omega)$ is the phase shift, c is the light speed, and L is the thickness of the sample.²³ “

4) We modified all figures.